# Mercury evidence for combustion of organic-rich sediments during the end-Triassic crisis

Jun Shen [1✉], Runsheng Yin [2✉], Thomas J. Algeo[1,3,4], Henrik H. Svensen[5] & Shane D. Schoepfer[6]

The sources of isotopically light carbon released during the end-Triassic mass extinction remain in debate. Here, we use mercury (Hg) concentrations and isotopes from a pelagic Triassic–Jurassic boundary section (Katsuyama, Japan) to track changes in Hg cycling. Because of its location in the central Panthalassa, far from terrigenous runoff, Hg enrichments at Katsuyama record atmospheric Hg deposition. These enrichments are characterized by negative mass independent fractionation (MIF) of odd Hg isotopes, providing evidence of their derivation from terrestrial organic-rich sediments ($\Delta^{199}$Hg < 0‰) rather than from deep-Earth volcanic gases ($\Delta^{199}$Hg ~ 0‰). Our data thus provide evidence that combustion of sedimentary organic matter by igneous intrusions and/or wildfires played a significant role in the environmental perturbations accompanying the event. This process has a modern analog in anthropogenic combustion of fossil fuels from crustal reservoirs.

[1] State Key Laboratory of Geological Processes and Mineral Resources, China University of Geosciences, Wuhan, Hubei 430074, P.R. China. [2] State Key Laboratory of Ore Deposit Geochemistry, Institute of Geochemistry, Chinese Academy of Sciences, Guiyang, Guizhou 550081, P.R. China. [3] State Key Laboratory of Biogeology and Environmental Geology, China University of Geosciences, Wuhan, Hubei 430074, P.R. China. [4] Department of Geology, University of Cincinnati, Cincinnati, OH 45221-0013, USA. [5] Centre for Earth Evolution and Dynamics (CEED), University of Oslo, Oslo, Norway. [6] Department of Geoscience and Natural Resources, Western Carolina University, Cullowhee, NC 28723, USA. ✉email: shenjun@cug.edu.cn; yinrunsheng@mail.gyig.ac.cn

Global environmental and biotic perturbations, including extreme warming, ocean acidification and anoxia, and a mass extinction, characterize the Triassic–Jurassic (T–J) transition (~201 Ma)[1–4]. These consequences are believed to have been driven by rising atmospheric $CO_2$ levels, which have been linked to the emplacement of the Central Atlantic Magmatic Providence (CAMP). Although the magnitude of the $CO_2$ rise (~2–4×) during the T–J transition has been estimated by plant stomatal ratios as well as pedogenic carbonate isotopes[5,6], debate continues regarding the source(s) of excess $CO_2$, with proposals of mantle-derived $CO_2$[7], heating and devolatilization of organic-rich and carbonate-bearing sedimentary rocks[8], and release of methane from permafrost or marine clathrates[9]. These hypotheses are partly motivated and supported by the negative carbon isotope excursions (CIEs) observed in both organic and inorganic records of the T–J transition. For example, a direct link between the CAMP and $CO_2$ release from organic-rich sediments in Brazilian basins was recently inferred by Heimdal et al.[8] based on carbon-cycle modeling combined with radiometric dating constraints. However, the negative CIE of the T–J transition ranges in magnitude from <1‰ to ~8‰ in different settings, suggesting that biological and chemical processes, preservation, as well as sedimentation rates imparted a large local overprint on the global signal[10], which complicates analysis of the magnitude of CAMP emissions based on carbon-cycle mass-balance calculations[11].

Mercury (Hg) concentrations and isotope ratios in sediments are widely used as a proxy for volcanic activity in stratigraphic successions (see review by Grasby et al.[12], Supplementary Notes 1–3). Volcanogenic emissions, the largest natural Hg source, supply Hg to the atmosphere with a residence time of 0.5–2 yr, playing a significant role in the global Hg cycle[13] (Supplementary Note 1). Owing to the low vapor pressure of Hg, massive volcanic inputs (e.g., from large igneous provinces, or LIPs) can overwhelm normal buffering mechanisms, leading to spikes in both raw and normalized Hg concentrations (e.g., ratios of Hg to total organic carbon, Hg/TOC) in diverse facies globally[14,15]. Moreover, Hg isotopes, especially mass independent fractionation (MIF) of odd isotopes (i.e., $\Delta^{199}Hg$), yield characteristic values in different reservoirs, facilitating their use as a Hg provenance proxy[16] (Supplementary Note 1). MIFs of odd isotopes (i.e., $\Delta^{199}Hg$) are largely unaffected by physical, chemical, and biological processes other than photo-reduction[16]. Reservoir-specific MIF values can be used to interpret Hg sources because $\Delta^{199}Hg$ values are near-zero for direct volcanic emissions from the deep Earth, distinguishing them from terrestrial and atmospheric fluxes, which generally show negative and positive $\Delta^{199}Hg$ values, respectively[16] (Supplementary Note 3). Paired measurement of Hg concentrations and isotopes thus greatly enhances the potential of Hg as a proxy for volcanism in Earth history.

Sedimentary Hg concentrations and isotopes around the T–J transition have previously been investigated in various marine, near-coastal and terrestrial settings (Fig. 1). Both marine and terrestrial sections show elevated Hg concentrations and Hg/TOC ratios during the T–J extinction interval, and these enrichments have been linked to volcanic sources based on near-zero $\Delta^{199}Hg$ values[17–19]. However, recent studies have painted a more nuanced picture, revealing that terrestrial and nearshore sections received Hg from multiple sources, including seawater, terrestrial materials (e.g., vegetation, soil), and the atmosphere[18–22]. Although processes of removal of Hg from pelagic seawater are complex[23,24], analysis of a pelagic open-ocean section far from continental influences is needed to gain a better understanding of atmospheric Hg fluxes during the T–J transition[15].

In this study, we present Hg concentration and isotope data (Supplementary Data 1) for a biostratigraphically well-dated pelagic radiolarian chert section from the central Panthalassic Ocean (Katsuyama, Japan; 35.4267°N, 136.9591°E, see Methods). The remote oceanic location of the study section, far from any continental margin or known volcanic arc, strongly limits the potential for terrestrial or non-CAMP volcanic influences[25], increasing the potential for recovering a CAMP signal. Furthermore, owing to slow rates of pelagic sedimentation at abyssal water depths, the study section has the potential to yield a more globally integrated signal of atmospheric Hg transport than most other T–J transitional sections analyzed to date. Finally, the location of the study site antipodally to the CAMP allows an assessment of the areal extent of CAMP volcanic influences.

## Results

**Geochemical records**. Hg concentrations are mainly low (<5 ppb) in the background intervals below and above the T–J extinction interval (−3.2 m to −2.0 m) but rise to >60 ppb (max 163 ppb) within it (Fig. 2a). Total organic carbon (TOC) and total sulfur (TS) values are low (<0.2%) throughout the section, without any significant variations between the background and extinction intervals (Fig. 2b, c). Similarly, thorium (Th) concentrations exhibit limited variations (<3%) throughout the section, with two peaks at −3.2 m (13.2%) and 0.8 m (6.5%) (Fig. 2d). Ratios of mercury to total organic carbon (Hg/TOC), total sulfur (Hg/TS), and thorium (Hg/Th) rise to >500 ppb/%, >2000 ppb/%, and >20 ppb/% in the T–J extinction interval from the background values of <50 ppb/%, <500 ppb/%, <5 ppb/%, respectively (Fig. 2b–d). Mass independent fractionation (MIF) of odd Hg isotopes ($\Delta^{199}Hg$) shows slightly positive values (0 to +0.11‰) within the background intervals and slightly negative values (−0.14 to −0.05‰) within the extinction interval (Fig. 2e). Mass independent fractionation (MIF) of even Hg isotopes ($\Delta^{200}Hg$) exhibits near-zero values within the background intervals and slightly negative values (−0.08 to 0.06‰) within the extinction interval (Fig. 2f).

## Discussion

Mercury has been used in T–J transitional research to track mantle-derived volcanic inputs to continental as well as shallow- to intermediate-depth marine sections proximal to the Pangean supercontinent[11,14,17–19,26,27] (Fig. 1). Near-zero $\Delta^{199}Hg$ values in association with Hg enrichments have been reported from two continental[19] and three shelf-slope sections[17,18]. This relationship was interpreted as evidence of volcanogenic Hg inputs linked to the CAMP, although in nearshore settings an intermediate $\Delta^{199}Hg$ value (i.e., near-zero) could also be produced through mixing of multiple Hg sources, e.g., a combination of terrestrial (negative $\Delta^{199}Hg$) and atmospheric (positive $\Delta^{199}Hg$) inputs[18–22]. Mercury data from a remote pelagic setting such as Katsuyama permit the global atmospheric signal to be isolated more effectively.

At Katsuyama, the T–J extinction interval is characterized by elevated raw (Fig. 2a) and normalized (TOC-, TS-, Th-, Fig. 2b, c, d) Hg concentrations, providing evidence of excess Hg loading. The average excess loading exceeds background values by a factor of 57X for raw Hg (Fig. 2a), 71X for Hg/TOC (Fig. 2b), 60X for Hg/TS (Fig. 2c), and 49X for Hg/Th (Fig. 2d). This pattern cannot be attributed to local controls[12,28–30], e.g., sedimentation rates, redox conditions, or specific host minerals (Supplementary Note 2), an inference supported by the global distribution of Hg-enriched T–J transition sections (Fig. 1). The excess Hg at Katsuyama was almost certainly sourced from the atmosphere given the distance of this pelagic section from continents (i.e., several thousand kilometers). Although warm and humid climate conditions may have increased Hg weathering fluxes, most river-borne

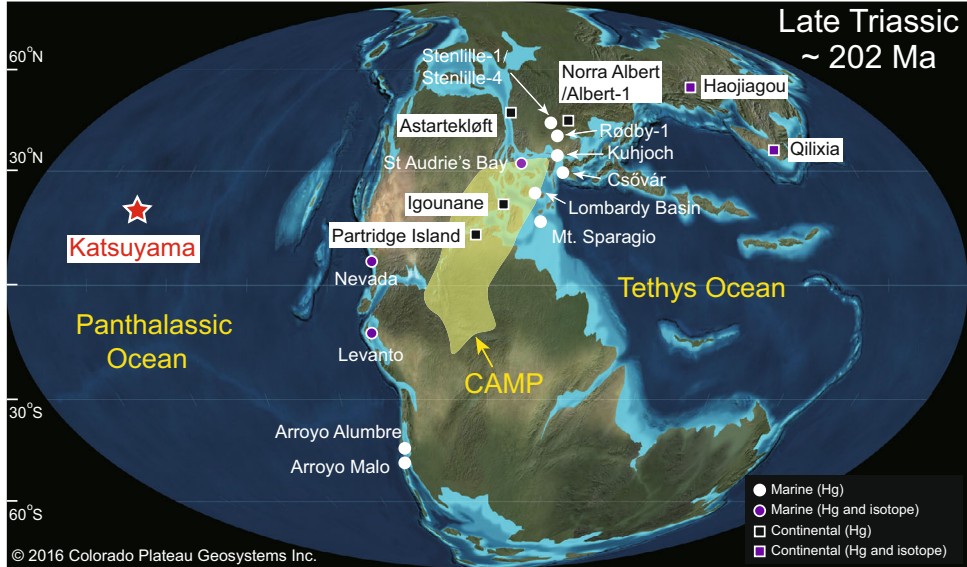

**Fig. 1 Late Triassic (~202 Ma) global paleogeography.** Geo-map adapted from Ron Blakey https://deeptimemaps.com/, (©2016 Colorado Plateau Geosystems Inc.) Red star shows the study site (Katsuyama section), located in the center of Panthalassa. Circles and squares represent other marine and continental sections, respectively, for which mercury data have been generated, including Hg concentrations and isotopes for Nevada[17,18], St Audrie's Bay[14,18], Levanto[18], Haojiagou, and Qilixia[19]; Hg concentrations for Arroyo Malo, Astartekløft, Igounane, Kuhjoch, Partridge Island (from ref. [14]); Stenlille-1/Stenlille-4, Rødby-1, and Norra Albert/Albert-1 (from ref. [26]), Csővár (from ref. [27]), Arroyo Alumbre (from ref. [11]), as well as Lombardy Basin, and Mt. Sparagio (from ref. [18]).

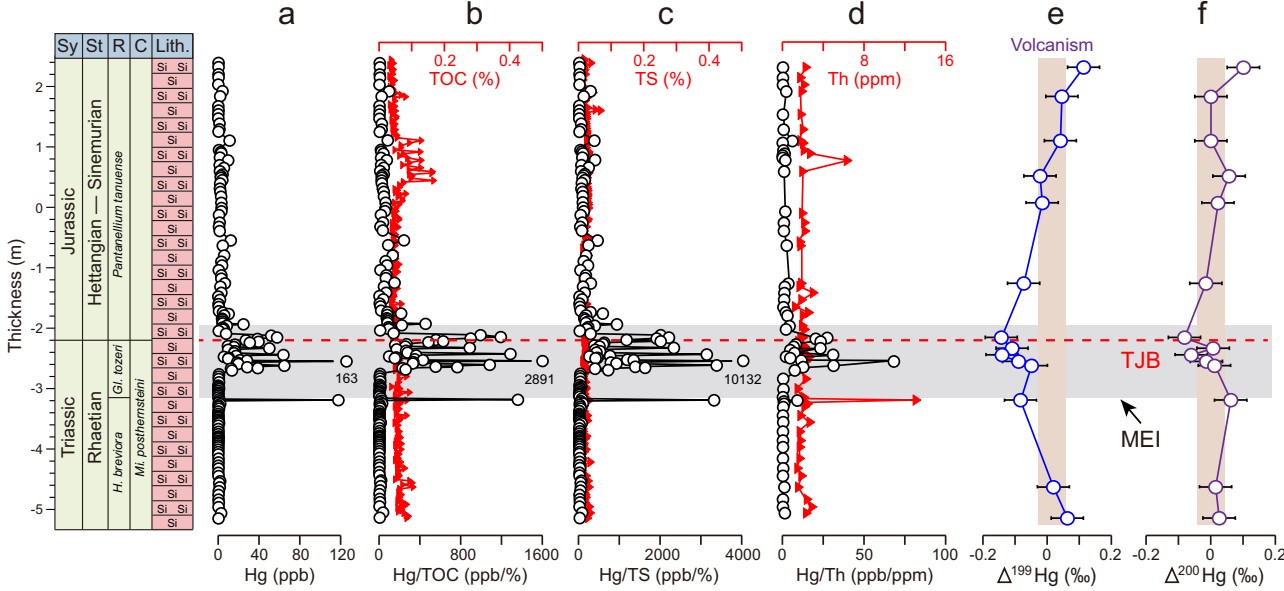

**Fig. 2 Chemostratigraphy of Katsuyama. a** mercury concentration (Hg, ppb); **b** total organic carbon (TOC, %, red triangle) and ratio of Hg to TOC (Hg/TOC, ppb/%, open circles); **c** total sulfur (TS, %, red triangle) and ratio of Hg to TS (Hg/TS, ppb/%, open circles); **d** thorium concentration (Th, ppm, red triangle) and ratio of Hg to Th (Hg/Th, ppb/ppm, open circles); **e** mass-independent fractionation of odd-Hg isotopes ($\Delta^{199}$Hg,‰); and **f** mass-independent fractionation of even-Hg isotopes ($\Delta^{200}$Hg,‰). The gray field represents the mercury-enriched interval (MEI), the vertical rectangle in e is volcanic $\Delta^{199}$Hg values (+0.02 ± 0.06‰[71]), and the vertical rectangle in f is $\Delta^{200}$Hg values of 0 ± 0.05‰[71]. Biozonation from references[40,59]. Horizontal bars in the $\Delta^{199}$Hg and $\Delta^{200}$Hg profiles are standard deviation (2σ) values. Abbreviations: Sy = System, St = (sub)stage, R = radiolarian zone, C = conodont zone; H. = Haeckelicyrtium, Gl. = Globolaxtorum, Mi. = Misikella. Source data are provided in the Supplementary data file.

detrital materials are deposited on continental shelves, within ~100 km of river mouths[31,32]. In addition to distance, a terrestrial Hg source is rendered unlikely due to lack of a significant correlation between Hg and Th (Supplementary Fig. 1), and to the relatively short residence time of Hg in seawater ($10^1$–$10^2$ years[33]), limiting its redistribution within the ocean system[32] (Supplementary Note 1). Furthermore, multiple Hg peaks near the extinction

interval in the present study section and at correlative sites[14,17–19] document that the CAMP eruptions were likely to have been pulsed, an inference supported by U-Pb dating[1,2].

Mercury isotopes, especially odd-isotope MIF (e.g., $\Delta^{199}$Hg), are a promising tool for tracking Hg sources to sediments (Supplementary Fig. 2 and Note 3). At Katsuyama, a significant perturbation of $\Delta^{199}$Hg (~2‰ decrease) is associated with T–J

transitional beds containing excess Hg, indicating a shift in the source(s) of Hg relative to the background flux ($\Delta^{199}$Hg > 0‰) (Figs. 2e, 3). Mercury readily participates in photochemical reactions during atmospheric transport, resulting in negative $\Delta^{199}$Hg in the gaseous Hg(0) species and positive $\Delta^{199}$Hg in the oxidized Hg(II) species[34]. For this reason, terrestrial Hg pools

normally show negative $\Delta^{199}$Hg due to the uptake of gaseous Hg(0)[35,36]. On the other hand, atmospheric Hg(II) is soluble and easily scavenged from the atmosphere through rainfall[37], resulting in oceanic Hg pools (e.g., seawater and marine sediments) showing positive $\Delta^{199}$Hg due to wet deposition of Hg[38] (Fig. 4). If Hg participates in photochemical reactions during atmospheric transport, more positive $\Delta^{199}$Hg values will be generated in remote settings than in proximal areas, although the magnitude of the MIF produced can vary[39].

Interpretation of Hg isotope signals in continental shelf settings is complicated by the influence of riverine terrestrial inputs. At Nevada, T–J transitional strata exhibit near-zero $\Delta^{199}$Hg values that were assumed to record volcanic release of deep-mantle Hg[17], although an alternative is mixing of atmospheric (positive MIF) and terrestrial (negative MIF) Hg sources. At Levanto[18], T–J transitional strata record only a slight decrease of $\Delta^{199}$Hg relative to the background flux (which was dominantly derived from seawater), rendering distinction of volcanic, atmospheric, and terrestrial sources difficult. At the epicontinental St. Audrie's Bay site[18], background Hg fluxes were dominantly of terrestrial origin ($\Delta^{199}$Hg −0.4‰ to −0.2‰), but rising $\Delta^{199}$Hg within the extinction interval (−0.17‰ to −0.07‰) resulted from either dominantly atmospheric deposition ($\Delta^{199}$Hg −0.17‰ to −0.07‰) or a mixture of atmospheric and terrestrial sources that were weighted toward the former. For the two terrestrial settings (Haojiagou and Qilixia)[19], the relatively higher $\Delta^{199}$Hg values near the extinction interval (−0.2‰ to −0.1‰, and −0.05‰ to +0.05‰ for Haojiagou and Qilixia, respectively) than that during the background interval (−0.4‰ to −0.2‰ for Haojiagou, and

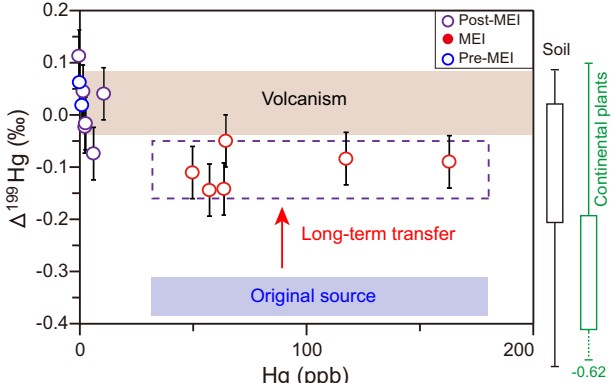

**Fig. 3 Crossplot of $\Delta^{199}$Hg versus Hg for Katsuyama.** The light blue rectangle represents the composition of the original source because $\Delta^{199}$Hg values become more positive at remote sites (purple dashed rectangle) due to atmospheric reactions[39]. Vertical bars for $\Delta^{199}$Hg represent standard deviation (2σ) values. The range of $\Delta^{199}$Hg values for volcanism, soil, and continental plants and sediments are from Yin et al.[71]. MEI = mercury-enriched interval. Source data are provided in the Supplementary data file.

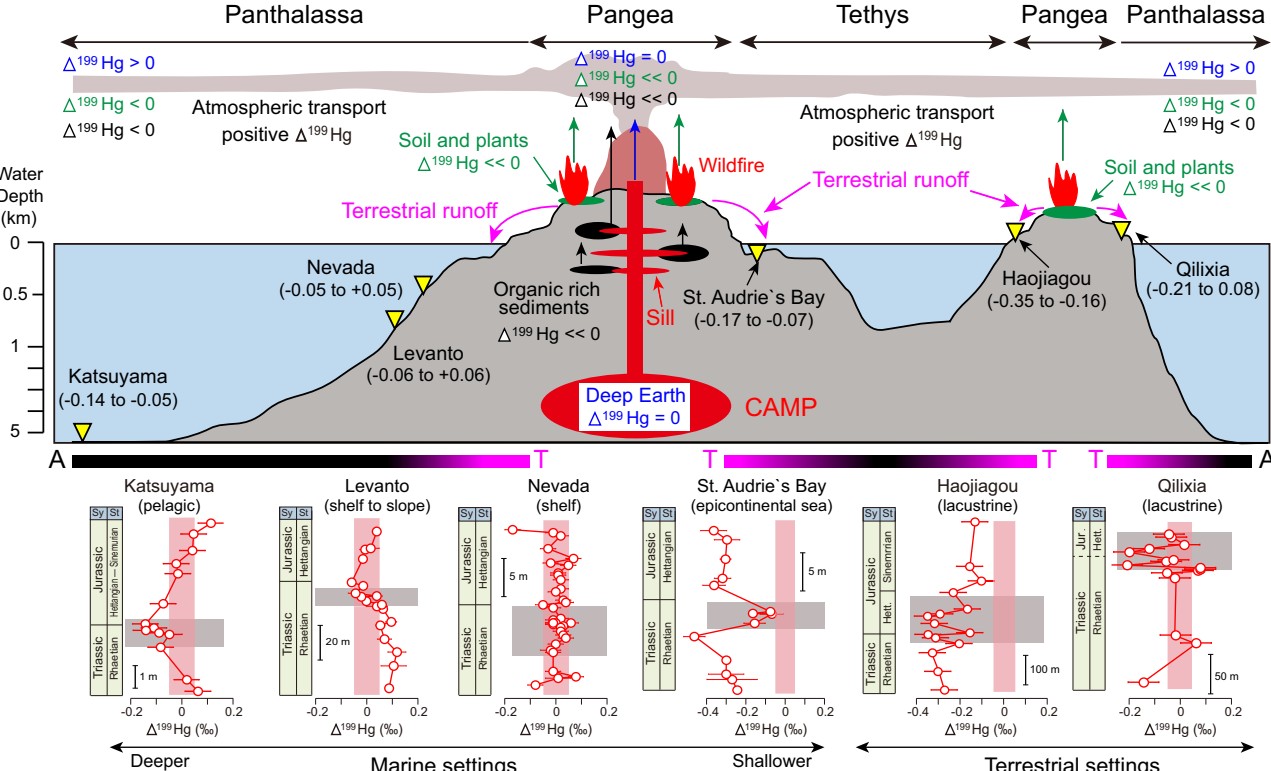

**Fig. 4 Schematic illustration of Hg cycle during the T–J transition.** Blue, green, and black vertical arrows represent Hg sourced from the deep Earth ($\Delta^{199}$Hg = 0‰), soil and plants (negative $\Delta^{199}$Hg), and organic-rich sediments (e.g., coal, negative $\Delta^{199}$Hg), respectively[16,71]. $\Delta^{199}$Hg > 0‰, <<0‰, and <0‰ represent positive, strongly negative, and weakly negative values of $\Delta^{199}$Hg. Numbers in brackets below the four sections represent $\Delta^{199}$Hg values of the MEI in units of ‰. The black and pink bars at bottom represent the atmospheric (A) and terrestrial (T) compositions of Hg at various sites in the Panthalassic and Tethys oceans. Horizontal bars of the $\Delta^{199}$Hg profiles represent standard deviation (2σ) values. Data sources by setting: pelagic (Katsuyama, this study); shelf to slope (Levanto[18], Nevada[17]), epicontinental sea (St. Audrie's Bay[18]), and terrestrial (Haojiagou and Qilixia[19]).

−0.2‰ to −0.05‰ for Qilixia) document mixing of terrestrial and atmosphere sources.

The negative $\Delta^{199}Hg$ values (ca. −0.14‰) associated with Hg spikes in the T–J extinction interval of the Katsuyama section have several potential sources, including photoreduction of Hg (II) complexed by reduced sulfur ligands in a euxinic photic zone, terrestrial inputs by soil erosion, and heating of organic matter in soils and sedimentary rocks (Fig. 4). Photic-zone euxinia is unlikely due to low concentrations of TS (Fig. 2c) and redox-sensitive trace elements[40] in the study section, and, thus, this process cannot account for the negative $\Delta^{199}Hg$ values[41]. Furthermore, an insignificant correlation between $\Delta^{199}Hg$ and TS also indicates limited influence by euxinia in both the photic zone and deeper water column (Supplementary Fig. 1). Terrestrial inputs of eroded soil through rivers are unlikely to have played a significant role given the thousands of kilometers separating Katsuyama from continental sources as well as the lack of correlations between Hg and Th[32]. We infer that heating of soil and terrestrial biomass (e.g., vegetation) through wildfires and/or contact metamorphism of organic-rich sedimentary rocks by igneous sills was the main source of low-MIF Hg.

Vegetation serves as a major sink of atmospheric Hg[13,42], because plants have a strong ability to accumulate gaseous Hg(0) through stomata, as shown by negative $\Delta^{199}Hg$ values in plant tissues[36,43]. Soil could accumulate a substantial amount of Hg from vegetation[13]. For this reason, terrestrial materials (e.g., vegetation and soil) are commonly characterized by elevated Hg concentrations[44] and negative $\Delta^{199}Hg$ values (e.g., ranging from −0.4‰ to 0‰[35,36,45]. Furthermore, negative $\Delta^{199}Hg$ values have been observed in coastal sediments that receive Hg mainly via terrestrial runoff[16,32]. Mercury has a close association with organic matter in organic-rich sediments, and decay of organic compounds can result in release of soil Hg to the atmosphere[13]. Negative $\Delta^{199}Hg$ values (−0.4 to −0.2‰) in the near-coastal, shallow-marine St. Audrie's Bay[18] and terrestrial Hajiagou[19] successions near the T–J transition are in agreement with results from similar present-day settings[16] (Fig. 4).

In the modern, wildfire combustion of vegetation and soil carbon is a major source of Hg to the atmosphere[46,47], and these sources are likely to have been important during the T–J transition as well. An increased frequency of wildfires linked to CAMP has been reported from many sections in Pangea as well as on the northern and eastern margins of the Tethys[3,48] (Supplementary Note 4). Wildfires cause combustion of plant debris and soil humic matter, releasing Hg with negative $\Delta^{199}Hg$ values to the atmosphere[16]. Alternatively, photoreduction of Hg(II) in the atmosphere can alter the isotopic signature of volcanic-Hg ($\Delta^{199}Hg$ near zero), resulting in negative $\Delta^{199}Hg$ values in the gaseous Hg(0) pool and positive $\Delta^{199}Hg$ values in the aqueous Hg(II) species. Gaseous Hg(0) with negative $\Delta^{199}Hg$ that is taken up by land plants through their stomata can be subsequently released by wildfire combustion. However, this source is unlikely to have been important during the CAMP eruptions as pelagic sediments would have recorded $\Delta^{199}Hg$ values ranging only from near-zero (volcanic Hg inputs) to negative (wildfire-released Hg).

Another mechanism for release of Hg yielding negative $\Delta^{199}Hg$ values to the atmosphere is the subsurface combustion of organic matter in coal and other organic-rich sedimentary rocks (e.g., black shales) (Fig. 4). CAMP igneous sills, which had a collective volume of ~$10^6$ km$^3$, are widely distributed across South America, Africa, Europe, and North America. However, they are concentrated (~70%) in Brazilian basins[49] such as the Amazonas and Solimões, in which organic-rich sediments of the Barreirinha, Jandiatuba, Jaraqui and Ueré formations contain up to 8% TOC[50]. Heating of these formations by CAMP intrusives may have been a major contributor to elevated atmospheric $CO_2$

concentrations and the negative CIE through the T–J extinction interval[8,51]. Although Hg concentrations have not been reported for these organic-rich formations to date, Hg is generally about one order of magnitude higher in coals and black shales compared to other sedimentary rock types[44]. Contact metamorphism of organic-rich sediments, especially those containing abundant terrestrially derived organics, should therefore emit Hg exhibiting low MIF, potentially accounting for negative $\Delta^{199}Hg$ values in the T–J extinction interval at Katsuyama (Fig. 2). Contact combustion of organic-rich sediments is also evidenced by abundant methane-rich fluid inclusions trapped within late-stage magmatic quartz[52] and zircon crystals[53]. Similarly to the carbon[52], magmatic Hg concentrations and isotope values may have been altered by admixture of abundant isotopically light Hg from the heating of organic-rich sediments.

Further support for the hypothesis of mobilization of Hg from an organic-rich sedimentary source followed by atmospheric transport during the T–J transition is provided by negative excursions of $\Delta^{200}Hg$ (Fig. 2f). MIF of $^{200}Hg$ is produced exclusively in the atmosphere, e.g., through photo-oxidation of gaseous Hg(0) to Hg (II)[54]. Terrestrial soil erosion and volcanic emissions of mantle-sourced mercury cannot explain the coupled negative $\Delta^{199}Hg$ and $\Delta^{200}Hg$ excursions at Katsuyama because both of these sources are characterized by $\Delta^{200}Hg$ of ~0‰[35,55]. In contrast, vegetation (and organic-rich sediments derived therefrom) tend to show coupled negative $\Delta^{199}Hg$ and $\Delta^{200}Hg$ due to foliage uptake of Hg(0) from the atmosphere[16,36,45].

Heating of vegetation and soil in surface as well as organic-rich sediments (e.g., coal and black shale) in the subsurface represents a potentially larger Hg flux to the atmosphere than direct volcanic Hg release from the deep Earth. Sediments containing abundant volatiles such as Hg, carbon, sulfur, and halogens are likely to play a significant role in environmental and biotic perturbations[56,57]. Much evidence now supports the role of combustion of organic matter by intrusives (e.g., coal and black shales) and wildfires (e.g., vegetation and soil) as significant sources of Hg and carbon to the surface Earth during mass extinctions at the Permian–Triassic[15,20] and Triassic–Jurassic[8] boundaries. Combustion of fossil fuels was the largest source of carbon to the Earth's surface during 1850-2019, with cumulative emissions of 445 ± 20 GtC (gigatons of carbon)[58]. The average release rate was ~2.6 GtC yr$^{-1}$ over that interval, or one order of magnitude greater than the rate of release of carbon during the T–J transition (~0.21 GtC yr$^{-1}$)[8]. Thus, although it is hard to know the eruptive rates of CAMP, the amount and rate of release of greenhouse gases (e.g., $CO_2$) by modern mining and fossil fuel burning are larger than LIP-related perturbations in deep time, reflecting the seriousness of present-day anthropogenic emissions and their implications for the future climate state of the Earth.

In this work, elevated Hg concentrations within the Triassic–Jurassic transition of a pelagic section from the central Panthalassic Ocean provide evidence of excess Hg inputs via atmospheric loading. Negative MIF values associated with these Hg enrichments support an inference of thermogenic Hg generated through volatilization of organic matter in sedimentary rocks heated by CAMP-related igneous sills and/or in soils by wildfires. These processes are likely to have released large amounts of toxic gases that were harmful to the contemporaneous environment and biosphere. The release of Hg by heating of organic-rich sediments (analogous to modern anthropogenic fossil fuel combustion) is a significant factor in present-day biotic and environmental perturbations of the Earth-surface system.

## Methods

**Geological background and study section**. The Mino-Tamba terrane complex, running through central Japan, is one of the largest surviving areas of pelagic

sedimentation from prior to the Jurassic Period. These sediments, which were deposited several thousand kilometers from major sources of detrital sediment, accumulated at extremely low sedimentation rates and preserve a time-integrated signal of open-ocean conditions[59]. Our study section is part of the Mino Terrane, which runs from SW to NE across central Honshu. This accretionary complex is composed of late Paleozoic through Mesozoic igneous rocks, limestones, silici-clastic rocks, and bedded radiolarian cherts[60]. The cherts represent low-latitude pelagic deposits that accumulated below the oceanic calcite compensation depth[61,62]. The bedding of the sediments, which reflects variable chert content, was likely produced by variations in the sinking flux of siliceous radiolarian tests as well as climate-driven changes in the delivery of windblown silt and clay components[63–65].

The Katsuyama section is exposed on the northern bank of the Kiso River, approximately 5.5 km northeast of Inuyama, Aichi Prefecture, Japan (35.4228°N, 136.9708°E). The section is composed of bedded cherts, with individual cherts beds ranging in thickness from 1 to 10 cm, with most beds being <4 cm in thickness. The beds are separated by millimeter-scale shale partings and are generally red in color. There is no major facies change in the Katsuyama section across the Triassic–Jurassic boundary, though minor changes in the color of the chert may reflect variation in depositional conditions[64]. Beds dip nearly vertically and are deformed by a complex system of faults and tight folds mapped in detail by Fujisaki et al.[66]. The data presented in this study come from a single coherent structural block bounded by the F4 and F9 faults (Fujisaki et al.[66], their Fig. 2).

The lowermost beds of our measured section are ~3 meters (−5.1 m to −2.2 m) below the T–J boundary and are likely Rhaetian (latest Triassic) in age, based on the occurrence of the *Haeckelicyrtium breviora* and *Globolaxtorum tozeri* radiolarian assemblages as well as the *Misikella posthernsteini* conodont Zone[40]. The Triassic–Jurassic boundary (−2.2 m) in the Katsuyama section was located based on the transition from the *Globolaxtorum tozeri* radiolarian Zone to the *Pantanellium tanuense* Zone[59]. This placement is supported by the LAD of the conodont *Misikella posthernsteini*[59,62] and coincides with a noticeable change in the color of the chert[40,66]. The overlying Lower Jurassic beds (−2.2 m to 2.4 m) are assigned to the *Parahsuum tanuense* Zone[67,68] and exhibit a noticeable transition from predominantly red to predominantly gray cherts in the middle of Subzone III, indicating a Hettangian to Sinemurian (Early Jurassic) age for the change in environmental conditions[40,66].

**Sample preparation and elemental analyses**. Samples were trimmed to remove visible veins and weathered surfaces and pulverized to ~200 mesh in an agate mortar for geochemical analysis. Aliquots of each sample were prepared for various analytical procedures. Hg concentrations (n = 157) were analyzed using a Direct Mercury Analyzer (DMA80) at Yale University. About 150 mg for chert samples were used in this analysis. Results were calibrated to the Marine Sediment Reference Material MESS-3 (80 ppb Hg). One replicate sample and a standard were analyzed for every ten samples. Data quality was monitored via multiple analyses of MESS-3, yielding an analytical precision (2σ) of ±0.5% of reported Hg concentrations.

Carbon and sulfur concentrations (n = 259) were measured using an Eltra 2000 C–S analyzer at the University of Cincinnati. Data quality was monitored via multiple analyses of the USGS SDO-1 standard with an analytical precision (2σ) of ±2.5% of reported values for carbon and ±5% for sulfur. An aliquot of each sample was digested in 2 N HCl at 50 °C for 12 h to dissolve carbonate minerals, and the residue was analyzed for total organic carbon (TOC), with total inorganic carbon (TIC) obtained by difference.

Trace element abundances (n = 61) were measured by Agilent 7500a inductively coupled plasma mass spectrometry (ICP-MS) at the State Key Laboratory of Geological Processes and Mineral Resources, China University of Geosciences (Wuhan). About 50 mg of each sample powder were weighed into a Teflon bomb and then moistened with a few drops of ultra-pure water before addition of 1 mL HNO₃ and 1 mL HF. The sealed bomb was heated at 190 °C in an oven for more than 48 h. After cooling, the bomb was opened and evaporated at 115 °C to incipient dryness, then 1 mL HNO₃ was added and the sample was dried again. The resultant salt was re-dissolved with 3 mL 30% HNO₃ before it was again sealed and heated in the bomb at 190 °C for 12 h. The final solution was transferred to a polyethylene bottle and diluted in 2% HNO₃ to about 80 mL for ICP-MS analysis. Analysis of the international rock standards BHVO-2 and BCR-2 yielded an analytical precision better than 5%, according to the relative standard deviation (RSD).

**Mercury concentrations and isotopes**. A subset of samples (n = 14) was analyzed for Hg isotopes at the State Key Laboratory of Ore Deposit Geochemistry, Institute of Geochemistry, Chinese Academy of Sciences, Guiyang. A double-stage tube furnace coupled with 40% anti aqua regia (HNO₃/HCl = 2/1, v/v) trapping solutions was used for Hg preconcentration, prior to isotope analysis[69]. All the solutions were diluted to ~0.5 ng mL⁻¹ Hg in 10–20% (v/v) acids using 18.2 MΩ cm water, and analyzed by Neptune Plus multiple collector inductively coupled plasma mass spectrometer (Thermo Electron Corp, Bremen, Germany). The instrument was equipped with the HGX-200 system and an Aridus II Desolvating Nebulizer System (CETAC Technologies, USA) for Hg and Tl introduction, respectively.

NIST SRM 997 Tl standard (50 ng mL⁻¹) was used as an internal standard for simultaneous instrumental mass bias correction of Hg. The instrument was tuned using the NIST SRM 3133 Hg standard solution for maximum intensity of ²⁰²Hg signal using Ar gas flows, torch position, and lenses.

Hg isotopic results are expressed as delta (δ) values in units of per mille (‰) variation relative to the bracketed NIST 3133 Hg standard, as follows:

$$\delta^{202}\text{Hg} = [(^{202}\text{Hg}/^{198}\text{Hg})_{sample}/(^{202}\text{Hg}/^{198}\text{Hg})_{standard} - 1] \times 1000 \qquad (1)$$

Hg concentration and acid matrices in the bracketing NIST-3133 solutions were matched with neighboring samples. Any Hg-isotopic value that did not follow the theoretical mass-dependent fractionation (MDF) was considered an isotopic anomaly caused by mass-independent fractionation (MIF). MIF values, reported in Δ notation (Δ^{xxx}Hg), were calculated as the difference between measured δ^{xxx}Hg and the theoretically predicted δ^{xxx}Hg value, in units of per mille (‰), as follows:

$$\Delta^{xxx}\text{Hg} = \delta^{xxx}\text{Hg} - \beta \times \delta^{202}\text{Hg} \qquad (2)$$

where xxx = 199 or 200, and β is equal to 0.2520 and 0.5024 for ¹⁹⁹Hg and ²⁰⁰Hg, respectively[70]. NIST-3177 secondary standard solutions, diluted to 0.5 ng/mL Hg with 10% HCl, were measured every 10 samples. Standard reference material GSS-4 (soil) was prepared and measured in the same way as the samples. Analytical uncertainty was estimated based on replicate analyses of the NIST-3177 secondary standard solution and full procedural analyses of GSS-4.

## Data availability
The geochemical data generated in this study are provided in the Supplementary Data file.

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

## Acknowledgements

The authors are grateful to Ruiyang Sun for assistance with the lab works. This research was supported by Natural Science Foundation of China (92055201, 42072037) (J.S.), 111 Project from National Bureau of Foreign Experts and the Ministry of Education of China (BP0820004) (J.S.), and Centre of Excellence grant to CEED (223272) from the Norwegian Research Council (H.H.S.). This work is a contribution to IGCP Project 739.

## Author contributions

J.S. conceived the study and designed it with T.J.A., S.D.S., and R.S.Y.; J.S. performed mercury, carbon, sulfur, and trace element analyses; R.S.Y. analyzed Hg isotopes; J.S. and T.J.A. wrote the paper with significant input from R.S.Y., H.H.S., and S.D.S.

## Competing interests

The authors declare no competing interests.

**Additional information**

