## [Peer Review File · Nature Communications]

REVIEWER COMMENTS

Reviewer #1 (Remarks to the Author):

Dear authors

It was a pleasure reading your paper, which I find to be generally well-written and insightful. I think your results, interpretations and discussion is an excellent contribution to our knowledge on the CAMP volcanism and the end-Triassic mass extinction.

I have some questions and comments, minor revisions(see below). My main concerns are: some misunderstandings regarding references pertaining to wildfire activity in the manuscript and in TextS4; confusion regarding settings, e.g. the use of near-continental (an epicontinental sea is located on the continent, so are shelf settings).

I hope you will find my comments and suggestions useful when revising the paper.

Best regards

Sofie Lindström

Line 22–23. Normally, the extinction is referred to as the end-Triassic mass extinction, and although it is true that there are some extinctions also taking place in the earliest Jurassic, most of the extinctions occur prior to the first occurrence of *Psiloceras spelae*, the index taxon for the base of the Jurassic.

Line 40. “Many of these”...hm. Perhaps write: “These consequences are believed to have been driven by...”

Line 41–43. “constrained” is not the right word as none of these proxy methods are that accurate and also there are major discrepancies between their results. I suggest you use “estimated”.

Line 72. Near-continental settings – what is that? It’s not continental but near a continent? Makes no sense. Why not stick to various marine, near-coastal and terrestrial settings?

Line 77–78. This sentence is kind of biased since you are in fact presenting results from such a section. Just thinking pelagic open-ocean settings may have other complicating issues too, see Sanei et al. 2021 Scientific Reports.

Line 81. You cannot assess the global impact of the CAMP based on results from one locality. You can do so by comparing with all other records. I suggest you rewrite this sentence.

Line 82–83. This is good and provides a strong argument.

Line 120. Add refs.

Line 124. With terrestrial here, you mean through weathering? I am thinking that Hg in soot from wildfires can travel far and that is also terrestrial, so perhaps clarify what you mean.

Line 140-142. Up to now you have been discussing the hypothetical fractionation between sites depending on various factors. Therefore I think it is strange that you suddenly state: “In this case...will be...”. I think you should write “In which case,...should be...,” and then add “and the results from this and previous studies corroborate this.” Or something like that.

Line 152–154. Is actually heating of organic-rich sediments through wildfires the primary source of Hg released through wildfires? Do you mean soils instead of sediments? As far as I know most of the Hg is stored in leaf litter and incorporated in soils, etc.

Line 155–158. Yes, exactly my point. But how often do you actually burn coal during wildfires? I have never heard that. Peatlands, mires – yes!

Line 162. Omit “the” before organic-rich sediments.

Line 164. I would not characterize St. Audrie’s Bay as a coastal section. Besides coastal section refers more to the outcrop today. When deposited it was rather a in a predominantly shallow marine, near coastal setting. I think you need to clarify this.

Also this whole sentence is a bit strange. Why not write: “Negative...in the near coastal, shallow marine St. Audrie’s Bay succession are in agreement with results from similar present day settings”, or something like that. And please add a reference to the present day results.

Line 168. Yes, exactly, from soils – not sediment.

Line 168–170. Replace “increasing” with “increased”.

Here, I find it annoying that you haven't even bothered to refer to a single one of the papers dealing with increased wildfires at the TJB in the text. Just to clarify, as I can see that you have not referred to all of them correctly in Text S4 either:

Van De Schootbrugge et al. (2009) – PAH, some of which are probably pyrolytic and some due to coking of organic-rich sediments by magma intrusion.

Marynowski and Simoneit (2009) – pyrolytic PAH and macroscopic charcoal.

Belcher et al. (2010) - charcoal from E Greenland

Pienkowski et al. (2012) have used the data from the previous publication.

Petersen and Lindström (2012) focus on charcoal abundance and reflectance in coalseams and coaly shales from the Danish Basin (Denmark and Sweden). It does NOT contain any new data on PAH, but refers to and discusses the other papers.

Williford et al. (2014). Pyrolytic PAH from E Greenland.

Song et al. (2020). Pyrolytic PAH from China.

All of these papers were discussed in the review by Lindström et al. (2020).

Please make sure to correct the misinformation in Text S4 and refer to at least one of the papers in the text.

Line 178. "having organic-rich sediments with TOC up to 8%".

Lines 185, and 187. "at the T-J"

Line 197–199. In line with comments above, explain this better. Separate the two processes. And perhaps you should write: "More and more evidence point to that combustion of organic-rich..." instead of ""is known".

Line 203–206. Actually we don't know this, because we can only estimate the emissions from CAMP in relation to the duration of the event as indicated by geochronology and magnetostratigraphy. But we have no idea how short or long the respective pulses of emissions really were and how much that was actually emitted annually from each pulse. Your statement unfortunately diminishes the importance of the CAMP volcanism and the end-Triassic crisis as an analogue for the severity of anthropogenic carbon emissions, even though you are stressing the severity of what we are seeing today.

In times like these, when funding agencies rather put their money on research on glaciology and what has been happening during the last 800kyrs - which only tells us what is happening but offers no clues to how the biota on Earth will cope or recover - than on research on deep time events that can teach us more about both the before, during and after scenarios of global warming events, it is probably not a good idea to downplay the importance of large igneous province events as analogues.

Line 300–302. You are five authors, but only four are mentioned here. How does that work?

Fig. 1. This figure is completely off. None of the localities is in the right place. I have seen this in other papers when Blakey's reconstructions are used as background, and I don't know if it is something that happens during the submission process, but please correct it!

Fig. 2. It is interesting that you also see some pulses in Hg loading just like in many of the other localities previously published.

Fig. 4. Here you show St. Audrie's Bay as located on the shelf of the Tethys Ocean. This is not correct as St. Audrie's Bay was located in the European epicontinental sea. My suggestion is that you extend the "shelf" area and add "epicontinental sea" above that and move Tethys further to the right. This can easily be achieved if you use smaller fonts.

Reviewer #2 (Remarks to the Author):

Review of: Mercury evidence for combustion of organic-rich sediments during the end-Triassic crisis

This paper presents a new Mercury concentration and isotopic dataset from a deep-water section that was in the middle of the Panthalassic Ocean during the end Triassic mass extinction. From this section, that contains a mercury isotope record unbiased by mixing with terrestrial sources, the authors show negative isotope anomalies and MIF. The data indicate that the mercury sources must have been highly negative since atmospheric interaction during mercury transport to the deep-water section changes the isotopic composition to more positive values, and since the record has negative values, the mercury source must have been much more negative. These data suggest a strong influence from mercury sources generated through the burning of vegetation in wildfires and the degassing of organically trapped mercury during interactions between CAMP magmas and organic rich sediments in the subsurface. It is argued that the influence of volcanically derived mercury during the CAMP eruptions may have been minor compared with mercury associated with the breakdown of organic matter.

The dataset presented here is very interesting and should be published, the paper is also well written (apart from some of the supplementary methods which contain numerous grammatical errors and should be read by a native English speaker). There are a few areas that could be explained

a little better, but in general the main text of the manuscript and the figures are in good shape. Some features of the data need more explanation, especially in regards to the published literature. I highlight some areas that the authors should discuss more below.

The first point is that the authors should indicate how they have located themselves in the Japanese section in the main text somewhere. At the moment its not clear how the authors are sure that they are looking at the correct interval before the reader gets to line 243.

There is some confusion about how the extinction interval is named in this study. It is commonly called the boundary interval, however the mass extinction occurs before the Triassic Jurassic boundary (see the age for the extinction in Blackburn et al. 2013, and the age for the boundary in Schoene et al. 2010 , updated in Wotzlaw et al. 2014). I would call this event the extinction interval, this interval overlaps with the Triassic Jurassic boundary, however the boundary should theoretically be a specific point in time (in reality its not quite, but theoretically it should be). This could be corrected throughout the text.

The data from the St Audries bay section deserve a bit more discussion in the main text, since they appear to show the opposite trend to the studied section and this should be explained in the main text. Its clear that the background levels in St Audries bay are relatively negative potentially due to run-off, however the extinction interval section changes towards more positive values. Could this be due to Hg input from magmatic sources?, Also the mercury isotopic composition in the Nevada section were previously used to argue for a volcanic source for this mercury (Thibodeau et al. 2016) – do you now dispute this conclusion? This should be discussed in the text. The mercury isotopic system is complex as is explained in the text, but if all of the records record different effects, how can we be sure of what the overall control is?

There has also been a suggestion that the CAMP lip may have been contaminated by organic rich sediments, promoting zircon crystallization (Davies et al. 2021). In this case, the Hg degassed from the magmas would be a mix of mantle derived Hg, and organic, sedimentary Hg, potentially changing the isotopic composition of the magmatically degassed Hg. This should also be discussed.

For the wildfire produced Hg enrichments. Is it possible that the flora at the time during the end Triassic acquired high Hg concentrations due to LIP degassing, and then were burnt in wildfires. Ultimately the source of the Hg would be the LIP, however it would have passed through organic matter changing its isotopic composition before being deposited in the studied section. Perhaps this is a complex pathway, but it is something that could be discussed. Considering that the LIP magmatism took place over 100's of Ka and the wildfires presumably occurred on much shorter time intervals.

Some smaller comments –

Line 77 – reference 19 is dealing with a different time period and doesn't seem appropriate here.

Line 189 – Mantle sourced carbon cannot explain... should be “mantle sourced mercury”

Line 195 – you mention that organic rich sediments represent larger Hg fluxes than LIP degassing, do you have modelling to show this? Do you know the magmatic flux of the CAMP or its initial Hg concentration (or the Hg concentration after contamination).

Line 250 – the wording is incorrect here, it seems like you are suggesting that the colour change is determining the age change, when the age change is independent of the colour change, just change the wording here and it should be ok.

Finally, the size of the figures is way too big. Its not possible to scroll down to see the figures and keep the text at a readable size (this is more a comment for your next publication..)

References cited –

Blackburn et al. 2013 – DOI - [10.1126/science.1234204](https://doi.org/10.1126/science.1234204)

Davies et al. 2021 – DOI - [10.1007/s00410-020-01765-2](https://doi.org/10.1007/s00410-020-01765-2)

Schoene et al. 2010 – DOI - [10.1130/G30683.1](https://doi.org/10.1130/G30683.1)

Wotzlaw et al. 2014 - DOI - [10.1130/G35612.1](https://doi.org/10.1130/G35612.1)

Point by point response to the comments of the editor and reviewers:

We thank the editor and two reviewers for comments which have led to significant improvements to our manuscript. We think we have adequately addressed all the comments from the reviewers and revised the manuscript accordingly. Please note that changes to the text are track-edited (see the WORD file named “main text with marked”), and the line numbers in our response letter refer to those of the revised manuscript.

REVIEWER COMMENTS

Reviewer #1 (Remarks to the Author):

Dear authors

It was a pleasure reading your paper, which I find to be generally well-written and insightful. I think your results, interpretations and discussion is an excellent contribution to our knowledge on the CAMP volcanism and the end-Triassic mass extinction.

I have some questions and comments, minor revisions(see below). My main concerns are: some misunderstandings regarding references pertaining to wildfire activity in the manuscript and in TextS4; confusion regarding settings, e.g. the use of near-continental (an epicontinental sea is located on the continent, so are shelf settings).

I hope you will find my comments and suggestions useful when revising the paper.

Best regards

Sofie Lindström

Reply: We are grateful for Prof. Sofie Lindström's positive comments. We made modifications as indicated in our responses below.

Line 22–23. Normally, the extinction is referred to as the end-Triassic mass extinction, and although it is true that there are some extinctions also taking place in the earliest Jurassic, most of the extinctions occur prior to the first occurrence of *Psiloceras spelae*, the index taxon for the base of the Jurassic.

Reply: Fair point. We modified the “Triassic-Jurassic mass extinction” to “end-Triassic mass extinction” in lines 22-23.

Line 40. “Many of these”...hm. Perhaps write: “These consequences are believed to have been driven by...”

Reply: Modified.

Line 41–43. “constrained” is not the right word as none of these proxy methods are that accurate and also there are major discrepancies between their results. I suggest you use “estimated”.

Reply: Fair point. We changed “constrained” to “estimated”.

Line 72. Near-continental settings – what is that? It’s not continental but near a continent? Makes no sense. Why not stick to various marine, near-coastal and terrestrial settings?

Reply: We meant the previous settings were around the continent, such as terrestrial, near-coastal, and shelf to slope settings as we stated in lines 106-108. We modified the sentence to “various marine, near-coastal and terrestrial settings” as suggested.

Line 77–78. This sentence is kind of biased since you are in fact presenting results from such a section. Just thinking pelagic open-ocean settings may have other complicating issues too, see Sanei et al. 2021 Scientific Reports.

Reply: Fair point. The processes of Hg loading in pelagic settings of the modern ocean are still largely unknown. However, it does not affect the main conclusions of the present study.

The elevated Hg enrichment near the TJB in the pelagic setting cannot be from terrestrial inputs by rivers (due to distance from continents), but it could result from the intense removal of Hg from seawater under suitable local conditions (e.g., strongly reducing bottomwaters). However, this is not the case for the present study section (Katsuyama): Firstly, we discussed possible controls on Hg loading (e.g., sedimentation rate, redox conditions, productivity) as well as the dominant host minerals of Hg in lines 118-131 and Supplementary Note 2; Secondly, the negative excursions of $\Delta^{199}\text{Hg}$ near the Hg-enriched interval show that Hg sources must be from terrestrial organic matter (with negative $\Delta^{199}\text{Hg}$ values, Blum et al., 2014) but not seawater or marine organisms (positive $\Delta^{199}\text{Hg}$ values, Blum et al., 2014) (lines 160-172).

Integration of Hg concentration and isotope data documents that the elevated Hg enrichments near the TJB at Katsuyama, an open-ocean section far from continental influences, were due to transfer of Hg through the atmosphere, which likely resulted from release of organic-hosted Hg from sediments/soils by magmatic intrusions and/or wildfires. We rephrased the sentence to indicate the complexity of Hg depositional processes in open-ocean settings and cited the two recent publications for the Hg removal of the modern pelagic settings (Sanei et al., 2021, SR; Liu et al., 2021, PNAS).

Line 81. You cannot assess the global impact of the CAMP based on results from one locality. You can do so by comparing with all other records. I suggest you rewrite this sentence.

Reply: Fair point. We removed the sentence.

Line 82–83. This is good and provides a strong argument.

Reply: Many thanks for the positive comments regarding this argument.

Line 120. Add refs.

Reply: Modified. We added Grasby et al., 2019 (ESR); Percival et al., 2018 (AJS) here.

Line 124. With terrestrial here, you mean through weathering? I am thinking that Hg in soot from wildfires can travel far and that is also terrestrial, so perhaps clarify what you mean.

Reply: Yes, we agree that Hg from burning of plants by wildfires also represents terrestrial Hg. We meant the terrestrial sources of Hg to the ocean linked to riverine inputs. We modified the statement here.

Line 140-142. Up to now you have been discussing the hypothetical fractionation between sites depending on various factors. Therefore I think it is strange that you suddenly state: “In this case...will be...”. I think you should write “In which case,...should be...,” and then add “and the results from this and previous studies corroborate this.” Or something like that.

Reply: Many thanks. We modified the sentence as suggested.

Line 152–154. Is actually heating of organic-rich sediments through wildfires the primary source of Hg released through wildfires? Do you mean soils instead of sediments? As far as I know most of the Hg is stored in leaf litter and incorporated in soils, etc.

Reply: In the modern system, wildfires release Hg to the atmosphere dominant from soils as well as from living and dead biomass (e.g., vegetation) (Biswas et al., 2007, Global Biogeochemical Cycles, Pirrone et al., 2010, Atmospheric Chemistry and Physics; Kumar et al., 2018, Atmospheric Environment). We rephrased the statements through the main text (lines 74, 170-171, 173-176, 184-186, 222, 227, 241) to make it clear.

Line 155–158. Yes, exactly my point. But how often do you actually burn coal during wildfires? I have never heard that. Peatlands, mires – yes!

Reply: Fair point. We rephrased it to “vegetation and soil” here.

Line 162. Omit “the” before organic-rich sediments.

Reply: Modified.

Line 164. I would not characterize St. Audrie's Bay as a coastal section. Besides coastal section refers more to the outcrop today. When deposited it was rather a in a predominantly shallow marine, near coastal setting. I think you need to clarify this.

Also this whole sentence is a bit strange. Why not write: "Negative...in the near coastal, shallow marine St. Audrie's Bay succession are in agreement with results from similar present day settings", or something like that. And please add a reference to the present day results.

Reply: Fair point. We agreed that St. Audrie's Bay represents deposition in a shallow marine environment. We modified the sentence as suggested (lines 181-183), and added a reference about the Hg isotope of present-day setting (line 183, Blum et al., 2014). In addition, we modified Figure 4 to clarify the depositional environment of each setting.

Line 168. Yes, exactly, from soils – not sediment.

Reply: Many thanks for the reminder. We modified it to vegetation and soil.

Line 168–170. Replace "increasing" with "increased".

Here, I find it annoying that you haven't even bothered to refer to a single one of the papers dealing with increased wildfires at the TJB in the text. Just to clarify, as I can see that you have not referred to all of them correctly in Text S4 either:

Van De Schootbrugge et al. (2009) – PAH, some of which are probably pyrolytic and some due to coking of organic-rich sediments by magma intrusion.

Marynowski and Simoneit (2009) – pyrolytic PAH and macroscopic charcoal.

Belcher et al. (2010) - charcoal from E Greenland

Pienkowski et al. (2012) have used the data from the previous publication.

Petersen and Lindström (2012) focus on charcoal abundance and reflectance in coalseams and coaly shales from the Danish Basin (Denmark and Sweden). It does NOT contain any new data on PAH, but refers to and discusses the other papers.

Williford et al. (2014). Pyrolytic PAH from E Greenland.

Song et al. (2020). Pyrolytic PAH from China.

All of these papers were discussed in the review by Lindström et al. (2020).

Please make sure to correct the misinformation in Text S4 and refer to at least one of the papers in the text.

Reply: Sorry for the missing wildfire references in the main text. We made modifications as follows: 1) We changed "increasing" to "increased"; 2) We added two references (Petersen and Lindström, 2012, Plos One; Lindström et al., 2021, ESR) here rather than citations to all the relatively papers in the comment above owing to the reference limit for NC; 3) We rephrased the Supplementary Note 4, and cited all the references in the supplementary information.

Line 178. “having organic-rich sediments with TOC up to 8%”.

Reply: Modified.

Lines 185, and 187. “at the T-J”

Reply: Modified.

Line 197–199. In line with comments above, explain this better. Separate the two processes. And perhaps you should write: “More and more evidence point to that combustion of organic-rich...” instead of “is known”.

Reply: 1) We separated the two processes; 2) We rephrased the sentence as suggested.

Line 203–206. Actually we don’t know this, because we can only estimate the emissions from CAMP in relation to the duration of the event as indicated by geochronology and magnetostratigraphy. But we have no idea how short or long the respective pulses of emissions really were and how much that was actually emitted annually from each pulse. Your statement unfortunately diminishes the importance of the CAMP volcanism and the end-Triassic crisis as an analogue for the severity of anthropogenic carbon emissions, even though you are stressing the severity of what we are seeing today.

In times like these, when funding agencies rather put their money on research on glaciology and what has been happening during the last 800kyrs - which only tells us what is happening but offers no clues to how the biota on Earth will cope or recover - than on research on deep time events that can teach us more about both the before, during and after scenarios of global warming events, it is probably not a good idea to downplay the importance of large igneous province events as analogues.

Reply: Good points. We fully agree that the duration of CAMP and the carbon release rate are hard to know due to uncertainties of dating of the sediments. In fact, we did not mean to downplay the importance of LIPs. We intended merely to make a comparison of scale between CO₂ emissions of the TJB event and those of the modern world. We modified the statements to address this point in lines 232-236.

Line 300–302. You are five authors, but only four are mentioned here. How does that work?

Reply: Modified. We changed “J.S. and T.J.A. wrote the paper with significant inputs from other co-authors” to “J.S. and T.J.A. wrote the paper with significant inputs from R.S.Y., H.H.S., and S.D.S” to make clear about the contributions from all the five authors (lines 322-325).

Fig. 1. This figure is completely off. None of the localities is in the right place. I am have

seen this in other papers when Blakey's reconstructions are used as background, and I don't know if it is something that happens during the submission process, but please correct it!

Reply: We do not know why everything except the paleomap was misplaced in the merged PDF file (in fact, we did not see any misplacements in our system), which is likely to have been due to an error during the PDF merging process. We will save the figures as PDF files in the next submission.

Fig. 2. It is interesting that you also see some pulses in Hg loading just like in many of the other localities previously published.

Reply: Yes, there were likely multiple pulses of CAMP eruptions. Similar patterns of Hg peaks in various settings may eventually permit a reconstruction of the detailed eruption history of CAMP volcanism. We stated this in lines 129-131.

Fig. 4. Here you show St. Audrie's Bay as located on the shelf of the Tethys Ocean. This is not correct as St. Audrie's Bay was located in the European epicontinental sea. My suggestion is that you extend the "shelf" area and add "epicontinental sea" above that and move Tethys further to the right. This can easily be achieved if you use smaller fonts.

Reply: Fair point. We modified the depositional environment of St. Audrie's Bay by extend the "shelf" area. We did not move the "Tethys" to the right for the European epicontinental sea also belong to the Tethys Ocean. We added the depositional environment under the section name. Furthermore, we added two terrestrial section (Haojiagou and Qilixia) sections in the new version of figure 4.

Reviewer #2 (Remarks to the Author):

Review of: Mercury evidence for combustion of organic-rich sediments during the end-Triassic crisis

This paper presents a new Mercury concentration and isotopic dataset from a deep-water section that was in the middle of the Panthalassic Ocean during the end Triassic mass extinction. From this section, that contains a mercury isotope record unbiased by mixing with terrestrial sources, the authors show negative isotope anomalies and MIF. The data indicate that the mercury sources must have been highly negative since atmospheric interaction during mercury transport to the deep-water section changes the isotopic composition to more positive values, and since the record has negative values, the mercury source must have been much more negative. These data suggest a strong influence from mercury sources generated through the burning of vegetation in wildfires and the degassing of organically trapped mercury during interactions between CAMP magmas and organic rich sediments in the subsurface. It is argued that the the influence

of volcanically derived mercury during the CAMP eruptions may have been minor compared with mercury associated with the breakdown of organic matter.

The dataset presented here is very interesting and should be published, the paper is also well written (apart from some of the supplementary methods which contain numerous grammatical errors and should be read by a native English speaker). There are a few areas that could be explained a little better, but in general the main text of the manuscript and the figures are in good shape. Some features of the data need more explanation, especially in regards to the published literature. I highlight some areas that the authors should discuss more below.

Reply: We are grateful for these positive comments. We made modifications as indicated in our responses below. Furthermore, our native English-speaking co-author Thomas Algeo has now read through and corrected the supplementary methods.

The first point is that the authors should indicate how they have located themselves in the Japanese section in the main text somewhere. At the moment its not clear how the authors are sure that they are looking at the correct interval before the reader gets to line 243.

Reply: Fair point. However, the NC policy request the Methods part behind the Conclusions other than the Introduction as others did. As response to the reviewer, we added the statements in the last paragraph of Introduction (lines 77-80) to make it clear.

There is some confusion about how the extinction interval is named in this study. It is commonly called the boundary interval, however the mass extinction occurs before the Triassic Jurassic boundary (see the age for the extinction in Blackburn et al. 2013, and the age for the boundary in Schoene et al. 2010, updated in Wotzlav et al. 2014). I would call this event the extinction interval, this interval overlaps with the Triassic Jurassic boundary, however the boundary should theoretically be a specific point in time (in reality its not quite, but theoretically it should be). This could be corrected throughout the text.

Reply: Fair point. We fully agreed with that 1) most of the extinction occurs in the Latest Triassic, yielding few extinction events in the earliest Jurassic; 2) the boundary should theoretically be a special point in time.

1) Follow the first reviewer`s suggestion, we named the extinction event as “end-Triassic mass extinction” other than “Triassic-Jurassic mass extinction” (lines 2, 22);

2) We changed “Triassic-Jurassic boundary interval” to “Triassic-Jurassic extinction interval” to refer the extinction event, which overlaps with the Triassic Jurassic boundary (lines 70-71, 91, 93, 97, 100, 102, 115, 129, 153, 156, 160-161, 204, 209);

3) we used “T-J transition” to present the interval from the Late Triassic to the Early

Triassic, which is more concise and sufficiently encompassing that it can include both the extinction interval and the boundary (lines 36, 39, 45, 48, 68, 77, 85, 106, 121, 134, 146, 149, 182, 185, 215, 232, 237, 539, 543);

4) We cited Blackburn et al., 2013 (lines 37, 131) and Schoene et al., 2010 (lines 37, 131) in the main text.

The data from the St Audries bay section deserve a bit more discussion in the main text, since they appear to show the opposite trend to the studied section and this should be explained in the main text. Its clear that the background levels in St Audries bay are relatively negative potentially due to run-off, however the extinction interval section changes towards more positive values. Could this be due to Hg input from magmatic sources?, Also the mercury isotopic composition in the Nevada section were previously used to argue for a volcanic source for this mercury (Thibodeau et al. 2016) – do you now dispute this conclusion? This should be discussed in the text. The mercury isotopic system is complex as is explained in the text, but if all of the records record different effects, how can we be sure of what the overall control is?

Reply: Good point. Interpretation of Hg concentrations and Hg isotope records in sediments is complex, especially for terrestrial and shallow-marine settings.

First, for the St Audries` Bay section, the background Hg was dominantly terrestrially sourced because they have similar $\Delta^{199}\text{Hg}$ values (-0.2 ‰ to -0.4 ‰) to terrestrial materials. Although $\Delta^{199}\text{Hg}$ increased near the extinction interval compared to the background intervals (-0.2 ‰ to -0.4 ‰), the values of the $\Delta^{199}\text{Hg}$ are still negative (-0.17 ‰ to -0.07 ‰, Fig. 4) within the interval of elevated Hg, providing evidence of some change in Hg sources. It is hard to exclude terrestrial inputs from rivers in near-coastal shallow-marine settings such as St. Audrie`s Bay. These $\Delta^{199}\text{Hg}$ values could be due to: 1) terrestrial inputs from different materials, which have less negative $\Delta^{199}\text{Hg}$ values during the extinction interval compared to the background intervals; 2) dominantly atmospheric transfer of Hg from heating of organic matter by intrusives and/or wildfires because they have similar values to the extinction interval at Katsuyama (-0.14 to -0.05 ‰, Fig. 4); and 3) mixing of terrestrial inputs (negative MIF) and magmatic sources of Hg (near-zero MIF).

Second, this is similar to the Hg isotope record from Nevada, which shows near-zero values during the T-J extinction interval. This could be due to: 1) dominantly magmatic sources (near-zero MIF) (this is the conclusion in Thibodeau et al., 2016); or 2) mixing of marine sources (positive MIF) and terrestrial inputs from rivers (negative MIF). We didn't intend to dispute the conclusion by Thibodeau, but more caution is needed to explain the $\Delta^{199}\text{Hg}$ records of shallow-marine settings since rivers are the largest source of Hg to coastal oceans (Liu et al., 2021, NG).

Caution is also needed in interpreting Hg isotope records from terrestrial and nearshore settings. This is why the Katsuyama pelagic section is important and highly promising for

tracking Hg sources during the T-J transition—it is unbiased by mixing with terrestrial sources. We added statements to discuss the complexity of Hg isotope interpretations in various environments in lines 145-159.

There has also been a suggestion that the CAMP lip may have been contaminated by organic rich sediments, promoting zircon crystallization (Davies et al. 2021). In this case, the Hg degassed from the magmas would be a mix of mantle derived Hg, and organic, sedimentary Hg, potentially changing the isotopic composition of the magmatically degassed Hg. This should also be discussed.

Reply: Great point. Yes, it is possible that the Hg in magmas was a mixture of Hg from deep-mantle sources and Hg from combustion of sedimentary organic matter. However, this does not affect the conclusions of this study since the latter source is equivalent to Hg released directly from combustion of organic-rich sediments. We added statements in lines 207-213 to address this point. Besides, we added two papers about the evidences of contact combustion of organic-rich sediments by CAMP near the TJB (Capriolo et al., 2021; Davies et al., 2021).

For the wildfire produced Hg enrichments. Is it possible that the flora at the time during the end Triassic acquired high Hg concentrations due to LIP degassing, and then were burnt in wildfires. Ultimately the source of the Hg would be the LIP, however it would have passed through organic matter changing its isotopic composition before being deposited in the studied section. Perhaps this is a complex pathway, but it is something that could be discussed. Considering that the LIP magmatism took place over 100's of Ka and the wildfires presumably occurred on much shorter time intervals.

Reply: This is another great point. In fact, as we stated in lines 136-141, and supplementary Note 3, biotic and dark abiotic reactions do not produce significant MIF (i.e., $\Delta^{199}\text{Hg}$). In contrast, all photochemical reactions that have been studied to date produce changes in MIF. Plants did not change the Hg isotope when absorbing Hg, but photoreduction of Hg(II) in atmosphere alters the isotopic signature of Hg, resulting in negative $\Delta^{199}\text{Hg}$ values in the gaseous Hg(0) pool and positive $\Delta^{199}\text{Hg}$ values in the aqueous Hg(II) species (Berquist and Blum, 2007).

Although it's difficult to know the amount of Hg released from LIP eruptions in geological history, volcanic Hg from deep mantle has $\Delta^{199}\text{Hg}$ of close to zero (Zambardi et al., 2009). Once released to the environment, the isotopic signature of volcanic-derived Hg can be altered by Hg(II) photoreduction process, resulting in negative $\Delta^{199}\text{Hg}$ in the product Hg(0) that is readily uptaken by vegetation (Yin et al., 2013; Demers et al., 2013), and positive $\Delta^{199}\text{Hg}$ values in the residual Hg(II) species which is water-soluble and easily deposited to the ocean via wet deposition (Gratz et al., 2010; Chen et al., 2012). In modern environments, terrestrial soil show negative $\Delta^{199}\text{Hg}$ values due to vegetation uptake of gaseous Hg(0) followed by accumulation of vegetation Hg in soil (Biswas et al., 2008; Demers et al., 2013; Yin et al., 2013), marine sediments mainly show positive $\Delta^{199}\text{Hg}$

values due to predominantly receiving Hg(II) from seawaters (Yin et al., 2015; Strok et al., 2015).

For the present work, as the review raised, it could be the case that plants took up a large amount of Hg (0), which was produced by the photoreduction of volcanic-source Hg ($\Delta^{199}\text{Hg} = 0 \text{ ‰}$), and then releasing by wildfires. However, because Hg can be transported long distances through the atmosphere, if CAMP released a large amount of Hg from the deep mantle ($\Delta^{199}\text{Hg}$ near zero) that was then altered by photoreduction in the atmosphere, which result in negative and positive for Hg(0) and Hg(II) respectively. When the plants uptake a large amount of Hg (0), the Hg (II) signal should also have been recorded in pelagic marine sediments due to the short residence time of Hg in seawater (10^2 - 10^3 years, Selin, 2009). The Hg isotopic composition of marine sediments would then be near-zero and/or positive for some samples (direct CAMP sources) and negative for other samples (Hg released by wildfires) within the Hg enrichment interval. However, our $\Delta^{199}\text{Hg}$ values, which are negative (-0.14 to -0.05 ‰) during the Hg enrichment interval at Katsuyama, do not support this hypothesis. We added statements to discuss it in lines 189-196.

Some smaller comments –

Line 77 – reference 19 is dealing with a different time period and doesn't seem appropriate here.

Reply: Here, we cited the references about PTB (Grasby et al., 2019, *Geology*; Shen et al., 2019, *Geology*), and TOAE (Them et al., 2019, *EPSL*) to show the complex of Hg isotope records in nearshore settings. These papers proposed that the terrestrial runoffs are significant to the Hg depositions in the continental and shallow water settings, which would be also be similar to the Hg records during the TJB events. Besides, we added two papers (Yager et al., 2021, *ESR*, Shen et al., 2022, *NC*) to show the multiple source of Hg at the terrestrial and shallow-water settings near the TJB.

Line 189 – Mantle sourced carbon cannot explain... should be “mantle sourced mercury”

Reply: Sorry for the error. Modified.

Line 195 – you mention that organic rich sediments represent larger Hg fluxes than LIP degassing, do you have modelling to show this? Do you know the magmatic flux of the CAMP or its initial Hg concentration (or the Hg concentration after contamination).

Reply: Good point. We have no idea about the Hg concentration of the direct releasing from deep-mantle by CAMP as well as the Hg concentrations in the organic-rich sediments in the South America, Africa, Europe, and North America basins, where the CAMP sills distributed. However, based on modern records, Hg is generally about one order of magnitude higher in coals and black shales compared to other sedimentary rock

types and volcanic direct releasing (Yudovich and Ketris, 2005, IJCG). Besides, the plants and soil are larger Hg reservoirs in the surface earth (e.g., Selin, 2009). The heating of organic-rich sediments by sills and wildfires are likely to be releasing a large amount of Hg to the atmosphere.

Besides, as we replied to the last major comment (in page 9-10 of this file), the $\Delta^{199}\text{Hg}$ records are promising to track the source of the elevated Hg. if we assumed the $\Delta^{199}\text{Hg}$ values are 0 for the deep-mantle source, which are similar to the modern volcanic records (Zambardi et al., 2009). Once released to the environment, the isotopic signature of volcanic-derived Hg can be altered by Hg(II) photoreduction process, resulting in negative $\Delta^{199}\text{Hg}$ in the product Hg(0) that is readily uptaken by vegetation (Yin et al., 2013; Demers et al., 2013), and positive $\Delta^{199}\text{Hg}$ values in the residual Hg(II) species which is water-soluble and easily deposited to the ocean via wet deposition (Gratz et al., 2010; Chen et al., 2012). The Hg isotopic composition of marine sediments would then be near-zero and/or positive for some samples (direct CAMP sources) and negative for other samples (Hg released by wildfires) within the Hg enrichment interval. However, our $\Delta^{199}\text{Hg}$ values, which are negative (-0.14 to -0.05 ‰) during the Hg enrichment interval at Katsuyama (in fact the original source would be more negative as shown in figure 3 and stated in lines 141-144), support that the direct releasing of volcanic-Hg paid insignificant role to the elevated Hg in sediments. We added statements to discuss it in lines 186-196.

Line 250 – the wording is incorrect here, it seems like you are suggesting that the colour change is determining the age change, when the age change is independent of the colour change, just change the wording here and it should be ok.

Reply: We rephrased parts of this paragraph to make it clear that, while we are commenting on the age of changes in chert color, these color changes are not serving as the basis for age interpretations.

Finally, the size of the figures is way too big. Its not possible to scroll down to see the figures and keep the text at a readable size (this is more a comment for your next publication..)

Reply: We are sorry for the inconvenience to read the figures. We do not know why the figures are too big in the merged PDF file. We saved the figures as PDF files for the revised submissions. Many thanks for the kindly reminder, we will pay more attention to the size of figures in the next publication.

References cited –

Blackburn et al. 2013 – DOI - 10.1126/science.1234204
Davies et al. 2021 – DOI - 10.1007/s00410-020-01765-2
Schoene et al. 2010 – DOI - 10.1130/G30683.1

Wotzlaw et al. 2014 - DOI - 10.1130/G35612.1

Reply: We cited Blackburn et al., 2013 in lines 37, 131; Davies et al., 2021 in line 211; Schoene et al., 2010 in lines 37,131. We did not cited Wotzlaw et al. (2014) for the references limit policy

REVIEWERS' COMMENTS

Reviewer #2 (Remarks to the Author):

I think the authors did an excellent job in dealing with the reviewers' comments. I don't have anything further to add. I think the manuscript represents a very nice piece of work that will be greatly appreciated in the literature.

I have no further comments.

Thanks very much for taking on board my comments in your final (I hope) version of the manuscript

Joshua Davies

Note: The reviewer had no comments to this version (see below).

REVIEWERS' COMMENTS

Reviewer #2 (Remarks to the Author):

I think the authors did an excellent job in dealing with the reviewers' comments. I don't have anything further to add. I think the manuscript represents a very nice piece of work that will be greatly appreciated in the literature.

I have no further comments.

Thanks very much for taking on board my comments in your final (I hope) version of the manuscript

Joshua Davies